# Functional Limitations and Use of General Health Examination and Cancer Screening Among People with Disabilities Who Need Support from Others: Secondary Data Analysis of the 2022 Comprehensive Survey of Living Conditions in Japan

**DOI:** 10.3390/ijerph22040484

**Published:** 2025-03-24

**Authors:** Takashi Saito, Kumiko Imahashi, Chikako Yamaki

**Affiliations:** 1Department of Social Rehabilitation, Research Institute of National Rehabilitation Center for Persons with Disabilities, 4-1 Namiki, Tokorozawa 359-8555, Japan; imahashi-kumiko@rehab.go.jp; 2Institute for Cancer Control, National Cancer Center, 5-1-1 Tsukiji, Chuo-ku 104-0045, Japan; cyamaki@ncc.go.jp

**Keywords:** Washington Group on Disability Statistics, functional limitation, disability, general health examination, cancer screening, disparity, Japan

## Abstract

Functional limitations that compromise preventive healthcare service utilization by people with disabilities in Japan are unelucidated. Secondary data from the Comprehensive Survey of Living Conditions (CSLC) in 2022 on the functional limitations defined by the Washington Group Short Set may bridge the knowledge gap, with evaluation of the generalizability of the knowledge as CSLC2022 was conducted in the aftermath of the COVID-19 pandemic. We described the number and types of functional limitations among people with disabilities who did not participate in the general health examination and cancer screenings. For the generalizability evaluation, we analyzed preventive service uptake and its relevance to disability status and compared them with compatible data from the CSLC from 2016, conducted before the pandemic. Among the eligible data, the percentage of people with disabilities among the preventive healthcare service non-participants was 2.40–3.66% (*n* = 129–239); one-third had two or more types of functional limitations. Functional limitations in mobility and self-care (basic activities of daily living [BADL]) are common and dominant issues. No obvious difference was observed regarding the aforementioned statistics between CSLC2022 and CSLC2016. Reasonable accommodation for people with BADL related to uni- or multifunctional limitations may contribute to increased accessibility to preventive healthcare services.

## 1. Introduction

Preventive health services, typically including general health examinations (GHEs) and cancer screening, are crucial for the early detection of medical conditions, prevention of illness, and maintenance of health [1]. Similarly to that in the general population, these services are important for people with disabilities to enable the early detection of secondary or comorbid health conditions and reduce the impact on health [2]. Preventive health services influenced the health-related outcomes of people with disabilities [3,4]. Despite these potential benefits, an increasing number of studies have documented that people with disabilities are disproportionately hindered from accessing preventive health services compared to their counterparts (people without disabilities) in both developed and developing countries [5,6,7,8,9,10,11]. This disability-based disparity in the access to preventive health services is a global public health issue [2].

The estimated number of people with disabilities in Japan is approximately eleven million six hundred thousand (constituting 9.2% of the total population) [12] and underscores the significance of scientific research focusing on potential disability-based disparities in use of preventive health services in Japan. Since around 1950 to 1960, the preventive health services, i.e., screening programs for cancers and general examinations for health conditions, started and generally spread in Japan [13]. Currently, general health examination and cancer screening are conducted nationwide as workplace-based or residential place-based programs under relevant laws [14]. The reported rates of participation in preventive health services among Japanese people with disabilities suggest that preventive health services might be less accessible for Japanese people with disabilities than for those without disability [15,16,17,18,19]. Although the results of these previous studies were informative, the generalizability of the evidence was limited because of the study samples—that is, convenience samples with a small number of study participants. Moreover, the participants of these previous studies were only people with disabilities, compromising the studies to compare between people with and without disabilities. Therefore, the complete picture of disability-based disparity and the associated challenges remains understudied in Japan. Therefore, to bridge the knowledge gaps regarding this health issue in Japan, studies that involve representative samples of people with and without disabilities and can provide reliable evidence are needed.

The Comprehensive Survey of Living Conditions (CSLC) is a nationwide, cross-sectional representative survey in Japan [20]. The survey included various questions on the living conditions of Japanese citizens. One part of the questions is on disability and the experience of preventive health service use: whether they need support from others in their daily life at the time of survey and participation or non-participation in the GHE and five types of cancer screenings in the past 1 or 2 years [20]. The CSLC, unfortunately, does not include specific questions regarding timing on the manifestation of disability or the administration of the GHE or cancer screenings. Although the lack of the information may consequently lead to some ambiguity of temporal sequence between current disability status and the previous use of the health service in the past 1 or 2 years, the CSLC data can provide an excellent opportunity for researchers to estimate the plausible causal effect of disability on the preventive health service use in Japanese individuals with disabilities. Therefore, the CSLC data can be considered a valuable source for analyzing preventive health service use among Japanese individuals with disabilities.

Saito et al. [11] analyzed secondary data from the CSLC conducted in 2016 (CSLC2016) and reported that, compared to the general population (43.7–73.4%), people with disabilities had lower rates of participation (20.8–50.0%). Moreover, after controlling for confounding factors, disability was identified as a significant factor for non-participation in the GHE and cancer screenings. Through a large-scale representative sample analysis, the results of this study provide an insight into the social issue of disability-based disparity in Japan. Although the previous study contributed new knowledge on the public health issues in Japan, some important knowledge gaps remain. Specifically, the data of CSLC2016 lacks information on the number and types of functional limitations. Thus, the specific functional limitations related to non-participation in preventive health services remain understudied. This knowledge would enable stakeholders (e.g., policymakers and service providers) to understand the specific characteristics of people with disabilities [21]: that is, those who need more support and what kinds of support would be helpful to mitigate the impact of the disability-based disparity on accessing preventive health services. Consequently, stakeholders can envision concrete and specific solutions to the issue. Knowledge of functional limitations related to service accessibility is indispensable for addressing this public health issue in Japan.

The CSLC conducted in 2022 (CSLC2022) was notable because the Washington Group Short Set (WGSS) [22], an internationally comparable disability measure, was for the first time adapted for use in the survey. The WGSS consists of six sub-questions on functional limitations: seeing, hearing, walking, climbing stairs, remembering or concentrating, self-care, and communication [22]. Although the WGSS also involves the ambiguity issue of temporal sequence with the previous use of the health service, analysis of data from the CSLC2022 could provide specific information on the functional limitations that may relate to nonparticipation in preventive health services and thereby bridge the knowledge gaps that persisted despite previous studies.

Although the analysis of data from CSLC2022 would be a good opportunity to add new knowledge, caution is needed when analyzing and interpreting them. The CSLC2022 was conducted in the aftermath of the COVID-19 pandemic. On 30 September 2021, the Japanese government lifted the state of emergency that had been declared during the COVID-19 pandemic [23]. However, a so-called semi-state of emergency, as a COVID-19 preventive measure to stop disease spread, continued until 21 March 2022 [23]. As the CSLC includes questions on an individual’s use of GHEs and cancer screenings in the past 1 or 2 years, their responses were based on their experiences during a state of full or semi-emergency in Japan. During these periods, except for immediately after the outbreak of COVID-19, the Japanese government encouraged local authorities to conduct the preventive health services as planned with careful consideration of individuals’ circumstances and precautionary measures to prevent the COVID-19 viruses from spreading at health facilities or health examination venues [24]. However, if these situations had drastically changed the behaviors toward preventive health service use among Japanese people with disabilities, the findings based on CSLC2022 would need to be considered specific to the pandemic situation. Although previous studies have reported the impacts of the COVID-19 pandemic on preventive health service use among the general population in Japan [25,26,27,28] and other countries [29,30], little is known about its impact on people with disabilities [31]. Therefore, the evaluation of the impact of COVID-19 needs to be accompanied by an analysis of CSLC2022 data to ascertain the generalizability of the findings. Considering the potential risks of new infectious disease pandemics in Japan [32,33], knowledge derived from this evaluation would contribute to the preparedness toward further pandemics.

The objectives of this study were twofold. The primary objective was to describe the number and types of functional limitations among people with disabilities who did not participate in a GHE or cancer screening. The secondary objective was to evaluate the impact of the COVID-19 pandemic on the generalizability of the findings related to the primary objective. For the evaluation, we utilized previously published compatible data based on CSLC2016 [11]. Data based on CSLC2016 and CSLC2022 were considered as representing the status of the pre-pandemic period and the aftermath of the pandemic, respectively. We analyzed preventive service uptake and its relevance to disability status based on the CSLC2022. Subsequently, the results of the analysis were compared with compatible data based on CSLC2016. We hypothesized that a functional limitation of mobility would be one of the main functional limitations, as previous studies [5,6] have shown that mobility issues are a primary barrier for accessing healthcare services, including preventive health services. Conversely, owing to the limited prior evidence that could be used to formulate a hypothesis, no hypothesis was developed for the secondary study objective because of its exploratory nature.

## 2. Materials and Methods

A cross-sectional study was conducted using secondary data from the CSLC2022 that were obtained on 5 February 2024, with permission from the MHLW. The statistical data presented in the current study were calculated by the authors based on unique inclusion/exclusion criteria and thereby might differ from publicly available data published by the Ministry of Health, Labor, and Welfare of Japan (MHLW).

The current study was approved by the ethics committee of the National Rehabilitation Center for Persons with Disabilities (approval no. 2024-069). The requirement for informed consent was waived because secondary data from the MHLW were used in the current study.

### 2.1. Data Source

The secondary data of the CSLC are provided to the researchers for research purposes with the permission of the MHLW. The Japanese government has conducted the CSLC as a nationwide, cross-sectional representative survey since 1986, with updated questions according to the times [20]. The CSLC consists of two types of surveys: large-scale surveys, conducted every 3 years, with larger sample sizes and wider scopes, and small-scale surveys, conducted annually, with smaller and narrower sample sizes [20]. The survey included various questions on the living conditions of Japanese citizens, such as demographic, health-related, and socioeconomic data [20]. The CSLC2022 was conducted as a large-scale survey.

The survey period for the CSLC2022 was from 2 June to 14 July 2022 [20]. The CSLC is a self-administered, nationwide, representative survey. In cases where it is difficult to answer a question owing to disease or disability, proxy responses are accepted. No reasonable accommodation for people with disabilities was included in the survey.

The CSLC2022 consists of four question modules: household, health, income and savings, and long-term care. For the current study, only the household, health, and income and saving modules were used for the analysis. Information on living conditions was included in the three modules as follows: demographic characteristics, health-related information, and socioeconomic status.

Random sampling methods were applied in CSLC2022, as in the previous CSLC. Detailed methods applied to CSLC2022 have been described previously [34]. Briefly, all household members from randomly selected 2000 census tracts (approximately 70,000 people from 30,000 households) were chosen as the study sample and asked to answer the questions in the household, health, and income and saving modules. Those who were in special circumstances, such as hospitalization or initialization for more than 3 months during the survey period, or foreigners who were unable to answer the questions written in Japanese, were exempted from answering all the questions. Questionnaires were collected personally, by mail, or online. The response rates of the household, health, income, and savings modules were 68.0%, 68.0%, and 61.2%, respectively [20].

### 2.2. Preventive Healthcare Service in Japan

The current study focused on the GHE and five types of cancer screening (lung, colorectal, gastric, cervical, and breast cancer), which are inquired in the CSLC.

In Japan, GHEs can be divided into two types: workplace-based and residential place-based programs [14]. Financial costs incurred, test items, frequency, and obligations depend on individuals’ circumstances. Typically, workplace-based programs are fully financed by employers and provided to all their employees, whereas residential place-based programs are mainly financed by local governments and provided to those who are self-employed or unemployed.

The five cancers are included for population-based cancer screening by the MHLW [35]. Furthermore, cancer screenings are provided as workplace-based or residential place-based programs with different financial burdens, depending on individual circumstances [36]. As of June 2022, when the CSLC2022 was conducted, the eligibility criteria for cancer screening were as follows: lung cancers via chest radiography for men and women aged ≥40 years annually; colorectal cancers via fecal immunochemical test for men and women aged ≥40 years annually; gastric cancers via upper gastrointestinal series (UGI) or endoscopy for men and women aged ≥50 years every 2 years, with UGI as an alternative for men and women aged ≥40 years annually; cervical cancers via Pap smear for women aged ≥20 years every 2 years; and breast cancers via mammography for women aged ≥40 years every 2 years [35]. The eligibility criteria for the upper age limit were not set by the Japanese government. Some residential place-based programs run by local governments, however, set upper age limitations, such as 74 or 79 years, owing to concerns over adverse incidents pertaining to screening [37].

### 2.3. Inclusion and Exclusion Criteria

To ensure comparability between the findings, the inclusion and exclusion criteria were the same as those used in a previous study [11]. The inclusion criterion was data from men and women aged 20–74 years. We excluded data from those who were exempt from answering the question under the aforementioned special circumstances and those with missing values for any variables.

### 2.4. Variables

We used the same variables as in a previous study [11] to ensure comparability. These variables were categorized for analysis similarly as in the previous study [11].

#### 2.4.1. General Health Examination and Cancer Screening

The following questions were asked about the experiences of GHE and cancer screening use: “Did you receive a GHE in the last year?” (answer options: “Yes” or “No”); “Did you undergo lung cancer screening (chest X-ray or sputum examination) in the last year?” (answer options: “Yes” or “No”); “Did you undergo colorectal cancer screening (fecal occult blood tests) in the last year?” (answer options: “Yes” or “No”); “Did you undergo gastric cancer screening (barium swallow test or endoscopic examination) in the last year?” (answer options: “Yes” or “No”); “Did you undergo breast cancer screening (mammography or breast ultrasound) in the last two years?” (Answer options: “Yes” or “No”); and “Did you undergo cervical cancer screening (Pap smear) in the last two years?” (answer options: “Yes” or “No”).

Regarding the question on gastric cancer screening, an additional question about the use of gastric cancer screening in the past 2 years is included in the CSLC2022 [38] but not in the CSLC2016. Thus, this additional question was not analyzed in the current study.

#### 2.4.2. Status of Disability

To ensure comparability with a previous study [11], the same question was used to define disability status: “Do you need any support or supervision from others because of your disability or declining physical function?” (answer options: “Yes” or “No”). The answer option “Yes” was considered as having a disability and “No” was considered as no disability.

#### 2.4.3. The Washington Group Short Set

The WGSS was used to define functional limitations. The six sub-questions of the WGSS were: “Do you have difficulty seeing, even if wearing glasses?”; “Do you have difficulty hearing, even if using a hearing aid?”; “Do you have difficulty walking or climbing steps?”; “Do you have difficulty remembering or concentrating?”; “Do you have difficulty with self-care, such as washing all over or dressing?”; and “Using your usual language, do you have difficulty communicating (e.g., understanding or being understood by others)?” Each sub-question had the same answer options: “no difficulty”, “some difficulty”, “a lot of difficulty”, or “cannot do at all”. Based on the recommendations of the Washington Group on Disability Statistics [22], answers of “a lot of difficulty” or ”cannot do at all” to any of the six sub-questions were considered to indicate functional difficulty. These criteria were applied to six individual sub-questions.

#### 2.4.4. Demographic, Physiological, and Psychosocial Variables

Twelve demographic, physiological, and psychosocial variables were used as confounding factors. The variables(categories) were: sex (men, women); age (20 to 39, 40 to 64, 65 and older); marital status (married, single, divorced/widowed); educational qualification (primary/junior high school, high school, vocational school/junior college/community (technical) college/university/post-graduate school); constant visits to hospitals or clinics, including for dentistry, acupuncture, moxibustion, Japanese massage or Judo therapy (Yes/No); subjective health status (good, normal, bad); alcohol consumption (never drank or quit drinking, social drinker/low-risk group (>0 to 100 g/week); middle-risk drinking (>100 to 350 g/week); high-risk drinking (>350 g/week); smoking habit (never/ex-smoker, current smoker); subjective financial state (wealthy, not poor and not wealthy, poor); Kessler Psychological Distress Scale (K6), a measure of mood and anxiety disorder [normal, total score 4), mild illness (5–12), severe illness (≥13)]; health insurance (National Health Insurance, employee insurance, other); employment status (employed, self-employed, employed (other), unemployed). Detailed information on these 12 variables has been provided in the previous report [11].

### 2.5. Statistical Analysis

All variables are presented as numbers and percentages. The statistical analysis was performed using IBM SPSS Statistics for Windows, Version 28.0. Armonk, NY: IBM Corp. The statistical significance level was set at *p* < 0.05.

First, we described the types and number of functional limitations among people with disabilities who did not participate in a GHE and cancer screening. Second, as a generalizability evaluation, the service uptake for GHEs and cancer screenings was calculated overall and separately among people with disabilities. The figure was then compared with compatible data from CSLC2016. Subsequently, binomial logistic regression analyses using the forced entry method were conducted to examine the relationship between disability status and preventive health service use. To ensure comparability, the analysis models were identical to those used in the previous study [11]. Disability status and preventive health service use were incorporated into the analysis model as explanatory and objective variables, respectively. Twelve confounding variables were incorporated into the model as confounding factors. Correlations between the independent and confounding variables were mild to moderate (phi coefficient and Cramer’s coefficient were 0.001–0.624).

## 3. Results

The data selection process is described in Figure 1. The original dataset was obtained from 45,160 study participants who answered questions on household, health, income, and savings modules. Data that met the exclusion criteria or had missing values were excluded from the analysis. Consequently, the eligible data for analysis were as follows: 23,868 men and women aged 20–74 for a GHE, 13,609 men and women aged 50–74 for gastric cancer screening, 18,091 men and women aged 40–74 for lung cancer screening, 18,132 men and women aged 40–74 for colorectal cancer screening, 9364 women aged 40–74 for breast cancer screening, and 12,165 women aged 20–74 for cervical cancer screening. Detailed information on the number of data points removed due to missing values is summarized and provided in the Appendix A).

Table 1 shows the characteristics of the data that were eligible for analysis. Disabilities, defined as the need for support or supervision from others, were observed in 443 (1.86%) participants. Regarding the six sub-questions of WGSS, the functional limitation of self-care was observed in 450 participants (1.89%) with the lowest prevalence among six WGSS sub-questions, and the functional limitation of mobility was observed in 818 participants (3.43%) with the highest prevalence among them.

Table 2 presents the disability status by participation or non-participation in preventive health services. The percentage of people with disabilities among the preventive health service participants was 0.80–1.36%. Conversely, the percentage of people with disabilities among the non-participants was 2.40–3.66, which was approximately three times larger. Among the total population, the overall participation rates in GHEs and lung, colorectal, gastric, cervical, and breast cancer screenings were 75.04% (17,911/23,868), 53.71% (9716/18,091), 48.47% (8788/18,132), 43.85% (5967/13,609), 43.72% (5318/12,165), and 47.01% (4402/9364), respectively. Similarly, the participation rates among people with disabilities were 50.79% (225/443), 34.22% (116/339), 29.91% (102/341), 26.38% (81/307), 22.64% (48/212), and 21.34% (35/164). The detailed characteristics of participants or nonparticipants in the GHEs and the five cancer screenings are presented in Appendix A.

Figure 2 shows the number of functional limitations defined by the WGSS among study participants with disabilities who did not participate in a GHE or the five types of cancer screenings. Overall, approximately 60% had at least one type of functional limitation, as defined by the WGSS, and one-third had two or more types of functional limitations (Figure 2). In contrast, approximately 40% of the participants had no functional limitations, as defined by the WGSS (Figure 2). Table 3 shows the number of individual functional limitations observed among participants with disabilities who did not participate in a GHE or the five types of cancer screenings. The total number of functional limitations and the number of participants for each preventive service group are inconsistent because the individual functional limitations are mutually exclusive. Overall, consistent patterns were observed across preventive health services; mobility and self-care were common and dominant functional limitations.

Figure 3 shows the participation rates of the GHE and five cancer screenings in CSLC2016 and CSLC2022. The CSLC2016 data were reproduced by Saito [11]. Overall, no obvious differences were observed between CSLC2016 and CSLC2022. Ranges of the rates among the total population in CSLC2016 and CSLC2022 were 43.7–73.4 and 43.7–75.0, respectively. The rates of CSLC2022 exceeded or were sustained in the GHE and four cancer screenings compared to those in CSLC2016. Similar tendencies were observed in the participation rates among people with disabilities. Specifically, the rates ranged from 20.8 to 50.0 and 21.3 to 50.8 in CSLC2016 and CSLC2022, respectively, resulting in slightly higher rates of CSLC2022 in the GHE and four cancer screenings than in CSLC2016.

Finally, Table 4 shows the results of the binomial logistic regression in CSLC2022 accompanied by the results in CSLC2016, which were reproduced from Saito et al. [11]. Disabilities were found to be a significant and independent contributing factor for non-participation in all preventive health services in the CSLC2022 after controlling for all confounding variables; the adjusted odds ratio (OR) was 1.34–2.10. These results were consistent with those of the CSLC2016, except for lung cancer screening, in which a non-significant association was observed between disability and non-participation. The detailed results, including the OR and adjusted OR of the 12 confounding variables, are shown in Appendix A.

## 4. Discussion

Our descriptive data analysis showed that more than half of the people with disabilities who did not participate in a GHE or cancer screening had at least one functional limitation that was defined by the WGSS, and one-third of the participants had two or more such limitations. Functional limitations in mobility and self-care were common and dominant issues across all types of preventive services. Moreover, no obvious changes in the uptake of preventive services and their relevance to disability status were observed between the pre-pandemic (CSLC2016) and aftermath of the pandemic (CSLC2022).

### 4.1. Preventive Health Service Use and the Types of Functional Limitations

Functional limitations regarding mobility and self-care were commonly observed among people with disabilities who did not participate in a GHE or cancer screening. As mobility and self-care are generally categorized as basic activities of daily living (BADL) [39], the BADL-related functional limitations were considered a dominant functional limitation that may relate to the use of preventive health services.

These findings are consistent with those of previous studies [5,6,40,41] that reported possible relationships between BADL related to functional limitations and preventive health service use. For instance, cross-sectional relationships between mobility impairment and lower utilization of preventive services were documented based on large-scale representative data from the UK [6] and the USA [5]. Moreover, functional limitations in dressing and transferring from one place to another, a BADL function, have been reported as potential barriers to preventive health services based on anecdotal and qualitative evidence [40,41].

Although our analyses did not provide a clear explanation of the potential causal mechanism between BADL-related functional limitations and the compromised use of preventive health services, the findings suggest that the BADL-related functional limitations may be a potential contributing factor toward compromising the use of preventive health services among people with disabilities. Reasonable accommodation for people with BADL related to functional limitations may contribute to making preventive health services more accessible.

Approximately 40% of people with disabilities who did not participate in a GHE or cancer screening had no functional limitation as defined by the WGSS. This implies that the six types of functional limitations defined by the WGSS did not fully cover functional limitations. One of the potential functional limitations may be related to psychological or mental conditions relating to functional limitation. This is because the WGSS is unable to capture mental-related issues because of the lack of a sub-question on the matter [22]. We previously confirmed that depression and other mental and musculoskeletal conditions related to functional limitations were not necessarily captured by the WGSS in the Japanese population [34]. Health conditions related to psychological or mental health may be related to the use of preventive health services.

### 4.2. Preventive Health Service Use and Multiple Functional Limitations

One-third of the people with disabilities who did not participate in a GHE or cancer screening had two or more types of the functional limitations defined by the WGSS. This finding highlights the importance of reasonable accommodations that can cater to individuals with multiple functional limitations. For instance, a woman with a disability who had functional limitations on mobility and vision due to diabetes mellitus would simultaneously need reasonable accommodations for mobility issues (e.g., removing physical barriers or providing wheelchairs) and vision issues (e.g., documents with large-sized print or braille). Individuals with multiple functional limitations have been reported as a potential subgroup of people with disabilities who are physically [42,43] and socially more vulnerable [44]. As multimorbidity, which may cause multiple functional limitations [45], increases with aging [46], multiple functional limitations and reasonable accommodation for those with them may need to be highlighted in the clinical practice of preventive health services in Japan, a super-aged society.

### 4.3. Generalizability of the Findings

Our findings are based on data from CSLC2022, which was conducted in the aftermath of the COVID-19 pandemic. This necessitates the authors to evaluate whether the findings can be applied when there is little or no influence of the pandemic.

Our comparison between two cross-sectional datasets obtained during the pre-pandemic period (CSLC2016) and in the aftermath of the pandemic (CSLC2022) showed no obvious change in the participation rate of preventive services. Moreover, independent relationships between disability status and non-participation in preventive services were observed in the aftermath of the pandemic, consistent with the relationships observed in the pre-pandemic period.

Our findings were inconsistent with those of previous studies that reported a notable decrease in the participation rate in preventive health services immediately after the COVID-19 pandemic stated [27,47]. However, Abubakar AK and colleagues [25] explored the impact on the uptake of breast cancer screenings among the general Japanese population in the aftermath of the COVID-19 pandemic (during 2021 to 2022) and reported that no clear decrease in the screening uptake was observed. Specifically, the uptake rate was 46.9% during this period, which Abubakar et al. considered to be no lower than the pre-pandemic uptake rate (38.2%) [25]. The findings of this current study are consistent with those of a previous study [25]. While it is beyond our scope to discuss possible explanations for the observed non-lower-uptake of preventive health services among people with disabilities, a possible link between vaccination uptake and increased breast cancer screening uptake was observed in the general Japanese population [25]. Considering that people with existing conditions, which may include people with disabilities, were prioritized for vaccination in Japan due to their vulnerability [48], the lack of lower uptake observed in the current study may, in part, relate to vaccination uptake among people with disabilities.

Ideally, data from a longitudinal cohort study that follows individuals with disabilities during the pre-pandemic period and in the aftermath of the pandemic should be used to evaluate the impact of the pandemic. However, the authors did not have data based on any longitudinal cohort studies. Therefore, we conducted an evaluation based on a comparison of two sets of cross-sectional data. Consequently, clear evidence suggesting that the research findings should not be applied when there is no or less influence of the pandemic was not found in the current study. Although the above-mentioned methodology-related limitations need to be considered, our evaluation suggested that our findings based on CSLC2022 might be applicable for a time when there is no or less influence of the pandemic.

### 4.4. Study Limitations

This study has several limitations that should be considered when interpreting the findings. First, our study findings were not able to show a clear causal relationship between disability, i.e., functional limitations, and the preventive health service use due to the ambiguity of the temporal sequence between them. To examine the causal relationship, further research using a longitudinal study design is needed. Second, because of the self-administered nature of the CSLC, some misclassifications of data may have occurred regarding the use of preventive health services, disability status, and WGSS. This misclassification may have influenced the results. Third, considering that no reasonable accommodations were provided during the CSLC, selection bias, which selectively omitted people with disabilities who had specific conditions (e.g., severe functional limitations), may have skewed our results. Moreover, proxy involvement due to a lack of reasonable accommodation may have influenced our results. Finally, as described above, our study was unable to examine the potential causal mechanism between functional limitations and the use of preventive health services. Further studies are required to explain why specific functional limitations can hinder access to preventive healthcare services, the barriers that hinder people with specific functional limitations, and how to solve these issues. This information can serve as an integral basis for developing concrete solutions to address this health issue.

## 5. Conclusions

To describe the number and types of functional limitations among people with disabilities who did not participate in a GHE and cancer screening, we analyzed data from the CSLC2022, which was conducted in the aftermath of the COVID-19 pandemic. Moreover, to evaluate the generalizability of the findings, the participation rate in preventive health services and the relationship between disability status and the use of preventive health services were compared before (CSLC2016) and in the aftermath of the pandemic (CSLC2022). Consequently, functional limitations regarding mobility and self-care (or when combined as BADL) were common and dominant issues. One-third of patients had two or more types of functional limitations. No obvious changes were observed between before and in the aftermath of the pandemic regarding the aforementioned statistics. Reasonable accommodation for people with BADL related to functional limitations and multiple functional limitations may contribute to making preventive health services more accessible. These findings appear to be applicable when there is little or no influence of the COVID-19 pandemic.

## Figures and Tables

**Figure 1 ijerph-22-00484-f001:**
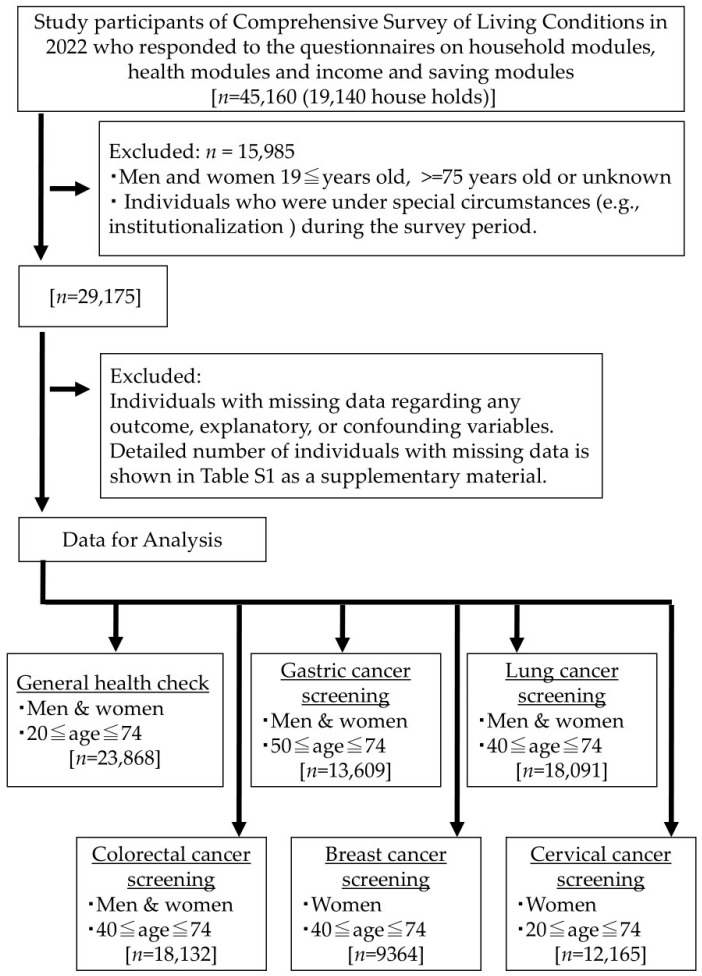
Data selection process.

**Figure 2 ijerph-22-00484-f002:**
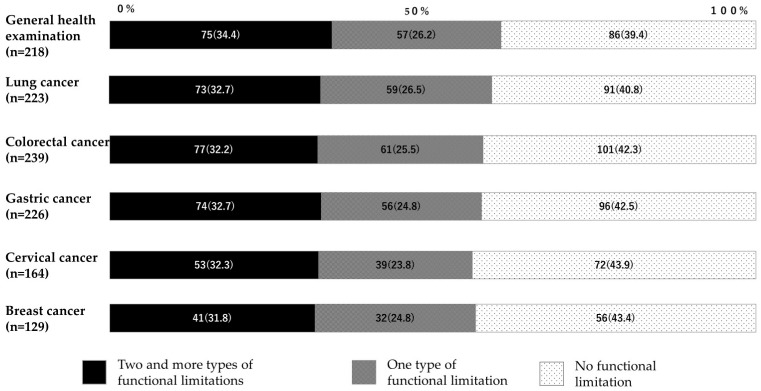
Number of functional limitations defined by The Washington Group Short Set among people with disabilities who did not participate in the general health examination or cancer screenings. [*n* (%)].

**Figure 3 ijerph-22-00484-f003:**
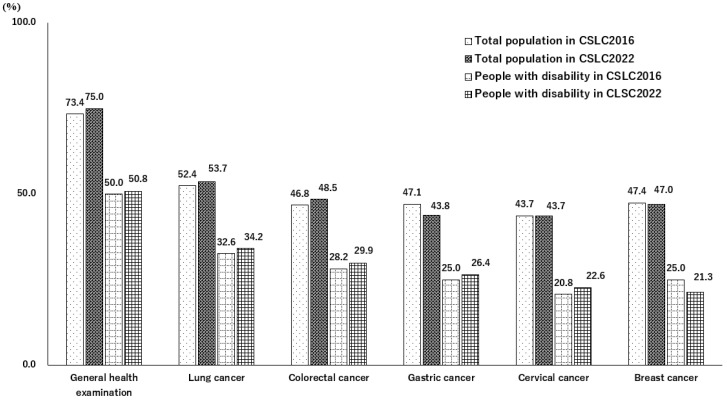
Participation rates of the general health examination and five cancer screenings among the total population and people with disabilities in the Comprehensive Survey of Living Conditions in 2016 and 2022. Note: Participation rates for the GHE and five cancer screenings in CSLC2016 and CSLC2022. The CSLC2016 data were reproduced from Sato et al. [11]. CSLC2016: Comprehensive Survey of Living Conditions conducted in 2016; CSLC2022: Comprehensive Survey of Living Conditions conducted in 2022.

**Table 1 ijerph-22-00484-t001:** Characteristics of study participants for analysis, both men and women aged 20–74 years (*n* = 23,868).

	Number	Percent
Disability		
No need for any support or supervision	23,425	98.14
Need for any support or supervision	443	1.86
WG sub-questions (vision)		
“No difficulty” or “some difficulty”	23,115	96.85
“A lot of difficulty” or “cannot do at all”	753	3.15
WG sub-questions (hearing)		
“No difficulty” or “some difficulty”	23,389	97.99
“A lot of difficulty” or “cannot do at all”	479	2.01
WG sub-questions (mobility)		
“No difficulty” or “some difficulty”	23,050	96.57
“A lot of difficulty” or “cannot do at all”	818	3.43
WG sub-questions (remembering or concentrating)		
“No difficulty” or “some difficulty”	23,307	97.65
“A lot of difficulty” or “cannot do at all”	561	2.35
WG sub-questions (self-care)		
“No difficulty” or “some difficulty”	23,418	98.11
“A lot of difficulty” or “cannot do at all”	450	1.89
WG sub-questions (communication)		
“No difficulty” or “some difficulty”	23,338	97.78
“A lot of difficulty” or “cannot do at all”	530	2.22
General health examination		
Participation	17,911	75.04
Non-participation	5957	24.96
Lung cancer screening ^(a)^		
Participation	9716	53.71
Non-participation	8375	46.29
Colorectal cancer screening ^(b)^		
Participation	8788	48.47
Non-participation	9344	51.53
Gastric cancer screening ^(c)^		
Participation	5967	43.85
Non-participation	7642	56.15
Cervical cancer screening ^(d)^		
Participation	5318	43.72
Non-participation	6847	56.28
Breast cancer screening ^(e)^		
Participation	4402	47.01
Non-participation	4962	52.99
Sex		
Male	11,576	48.50
Female	12,292	51.50
Age (years)		
65–74	6054	25.36
40–64	12,215	51.18
20–39	5599	23.46
Marital status		
Married	15,981	66.96
Single	5692	23.85
Divorced/widowed	2195	9.20
Educational qualification		
Vocational school/junior college/community (technical) college/university/post-graduate school	12,384	51.89
High school	10,053	42.12
Primary/junior high school	1431	6.00
Subjective financial state		
Wealthy	1470	6.16
Not poor, not wealthy	9876	41.38
Poor	12,522	52.46
Health insurance		
Employee insurance	16,098	67.45
National Health Insurance	7315	30.65
Other	455	1.91
Employment status		
Employed	14,367	60.19
Self-employed	1591	6.67
Employed (other)	1622	6.80
Unemployed	6288	26.34
Kessler Psychological Distress Scale		
Normal (total score ≦ 4)	17,895	74.97
Mild illness (5 ≦ total score ≦ 12)	5002	20.96
Severe illness (13 ≦ total score)	971	4.07
Constant visits to hospitals †		
Yes (constant visits)	11,060	46.34
No (no constant visits)	12,808	53.66
Subjective health status		
Good	9435	39.53
Normal	11,906	49.88
Bad	2527	10.59
Alcohol consumption		
Never or quit drinking	13,376	56.04
Social drinker/low-risk group (>0 to ≤100 g/week)	5320	22.29
Middle-risk drinking (>100 to ≤350 g/week)	4356	18.25
High-risk drinking (>350 g/week)	816	3.42
Smoking habit		
Never/ex-smoker	19,652	82.34
Current smoker	4216	17.66

† No clear definition of the frequency of visits was provided in the questionnaire. ^(a)^ Men and women aged 40–74 years (*n* = 18,091). ^(b)^ Men and women aged 40–74 years (*n* = 18,132). ^(c)^ Men and women aged 50–74 years (*n* = 13,609). ^(d)^ women aged between 20 and 74 years old (*n* = 12,165). ^(e)^ women aged between 40 and 74 years old (*n* = 9364). Note: WG, the Washington Group Short Set.

**Table 2 ijerph-22-00484-t002:** Participation and non-participation in the general health examination and cancer screenings stratified by disability status.

	General Health Examination (*n* = 23,868) ^(a)^ *n* (%) [95% CI of %]		Cancer Screening									
			Lung (*n* = 18,091) ^(b)^ *n* (%) [95% CI of %]		Colorectal (*n* = 18,132) ^(c)^ *n* (%) [95% CI of %]		Gastric (*n* = 13,609) ^(d)^ *n* (%) [95% CI of %]		Cervical (*n* = 12,165) ^(e)^ *n* (%) [95% CI of %]		Breast (*n* = 9364) ^(f)^ *n* (%) [95% CI of %]	
	Participants (*n* = 17,911)	Non Participant (*n* = 5957)	Participants (*n* = 9716)	Non-Participant (*n* = 8375)	Participants (*n* = 8788)	Non-Participant (*n*= 9344)	Participants (*n* = 5967)	Non-Participant (*n* = 7642)	Participants (*n* = 5318)	Non-Participant (*n* = 6847)	Participants (*n* = 4402)	Non-Participant (*n* = 4962)
Disability												
No need for any support or supervision	17,686 (98.74) [98.57, 98.90]	5739 (96.34) [95.84, 96.79]	9600 (98.81) [98.58, 99.01]	8152 (97.34) [96.98, 97.67]	8686 (98.84) [98.60, 99.05]	9105 (97.44) [97.11, 97.75]	5886 (98.64) [98.32, 98.91]	7416 (97.04) [96.64, 97.40]	5270 (99.10) [98.82, 99.33]	6683 (97.60) [97.22, 97.95]	4367 (99.20) [98.91, 99.44]	4833 (97.40) [96.93, 97.82]
Need for any support or supervision	225 (1.26) [1.10, 1.43]	218 (3.66) [3.21, 4.16]	116 (1.19) [0.99, 1.42]	223 (2.66) [2.33, 3.02]	102 (1.16) [0.95, 1.40]	239 (2.56) [2.25, 2.89]	81 (1.36) [1.09, 1.68]	226 (2.96) [2.60, 3.36]	48 (0.90) [0.67, 1.18]	164 (2.40) [2.05, 2.78]	35 (0.80) [0.56, 1.09]	129 (2.60) [2.18, 3.07]

^(a)^ Men and women aged 20–74 years. ^(b)^ Men and women aged 40–74 years. ^(c)^ Men and women aged 40–74 years. ^(d)^ Men and women aged 50–74 years. ^(e)^ Women aged 20–74 years. ^(f)^ Women aged 40–74 years. Note: CI, confidence interval.

**Table 3 ijerph-22-00484-t003:** Functional limitations observed among the participants with disabilities who did not participate in the general health examination or cancer screenings.

	General Health Examinations (*n* = 218)		Cancer Screening									
			Lung (*n* = 223)		Colorectal (*n* = 239)		Gastric (*n* = 226)		Cervical (*n* = 164)		Breast (*n* = 129)	
	*n*	(%)	*n*	(%)	*n*	(%)	*n*	(%)	*n*	(%)	*n*	(%)
Functional limitations												
Mobility	82	(28.98)	89	(33.33)	97	(35.14)	99	(36.94)	59	(29.21)	50	(31.85)
Self-care	55	(19.43)	58	(21.72)	56	(20.29)	58	(21.64)	42	(20.79)	33	(21.02)
Remembering or concentrating	50	(17.67)	45	(16.85)	44	(15.94)	40	(14.93)	34	(16.83)	28	(17.83)
Communication	57	(20.14)	40	(14.98)	40	(14.49)	33	(12.31)	40	(19.8)	24	(15.29)
Seeing	25	(8.83)	23	(8.61)	25	(9.06)	24	(8.96)	16	(7.92)	12	(7.64)
Hearing	14	(4.95)	12	(4.49)	14	(5.07)	14	(5.22)	11	(5.45)	10	(6.37)
(Total)	283	(100)	267	(100)	276	(100)	268	(100)	202	(100)	157	(100)

The total number of functional limitations and number of participants for each preventive service group are inconsistent because the individual functional limitations are mutually exclusive.

**Table 4 ijerph-22-00484-t004:** Unadjusted and adjusted odds ratio of disability for non-participation in the general health examination and cancer screenings.

	General Health Examination		Cancer Screening									
			Lung		Colorectal		Gastric		Cervical		Breast	
	Unadjusted OR [95% CI]	Adjusted OR [95% CI]	Unadjusted OR [95% CI]	Adjusted OR [95% CI]	Unadjusted OR [95% CI]	Adjusted OR [95% CI]	Unadjusted OR [95% CI]	Adjusted OR [95% CI]	Unadjusted OR [95% CI]	Adjusted OR [95% CI]	Unadjusted OR [95% CI]	Adjusted OR [95% CI]
CSLC2022												
No need for any support or supervision	1 [Ref]	1 [Ref]	1 [Ref]	1 [Ref]	1 [Ref]	1 [Ref]	1 [Ref]	1 [Ref]	1 [Ref]	1 [Ref]	1 [Ref]	1 [Ref]
Need for any support or supervision	2.99 [2.47, 3.61]	1.34 [1.09, 1.66]	2.26 [1.81, 2.84]	1.44 [1.13, 1.83]	2.24 [1.77, 2.82]	1.512 [1.18, 1.94]	2.21 [1.71, 2.86]	1.54 [1.17, 2.02]	2.69 [1.95, 3.72]	1.68 [1.19, 2.38]	3.33 [2.29, 4.85]	2.10 [1.42, 3.12]
CSLC2016												
No need for any support or supervision	1 [Ref]	1 [Ref]	1 [Ref]	1 [Ref]	1 [Ref]	1 [Ref]	1 [Ref]	1 [Ref]	1 [Ref]	1 [Ref]	1 [Ref]	1 [Ref]
Need for any support or supervision	2.81 [1.92, 4.11]	1.73 [1.14, 2.62]	2.30 [1.46, 3.62]	1.56 [0.96, 2.51]	2.26 [1.40, 3.64]	1.78 [1.08, 2.94]	2.71 [1.56, 4.70]	2.27 [1.27, 4.05]	3.00 [1.54, 5.85]	2.12 [1.04, 4.32]	2.74 [1.33, 5.62]	2.22 [1.04, 4.72]

Note OR: odds ratio; CI: confidence interval; Ref: reference. The adjusted OR and controlled confounding variables included sex, age, marital status, educational qualification, constant hospital visit, subjective health status, alcohol consumption, smoking habit, subjective financial state, Kessler Psychological Distress Scale, health insurance, and employment status. The data in CSLC2016 were reproduced from Saito [11].

## Data Availability

Data are contained within the article and Appendix A.

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
