# Peer review of "Functional Limitations and Use of General Health Examination and Cancer Screening Among People with Disabilities Who Need Support from Others: Secondary Data Analysis of the 2022 Comprehensive Survey of Living Conditions in Japan"

_ijerph, 2025, doi:10.3390/ijerph22040484_

Round 1

Reviewer 1 Report

Comments and Suggestions for Authors

I am honored to review your manuscript. The paper demonstrates  clear objectives. 

Recommendation: Reconsider after major revisions

A. Introduction

1-Need to Include a contextual paragraph describing Japan's healthcare system structure and its relevance to the study.

2-Condense the third paragraph regarding The Comprehensive Survey of Living Conditions (CSLC) and consider relocating relevant methodological details to the Methods section to avoid redundancy.

B-Methodological Concern: 

First-Cross-sectional Design and Temporality:

-The classification of this study as cross-sectional raises significant concerns regarding the temporal relationship between disability status and screening participation. The current design presents the following limitations:

1-Temporal Sequence:

-Disability status is assessed at the time of survey completion

-Screening participation is measured retrospectively (1-2 years prior)

-This creates potential temporal ambiguity in establishing cause-effect relationships

2-Disability Classification Issues:

-The study does not differentiate between permanent and temporary disabilities

-Examples of temporal ambiguity:

-A respondent with a current leg fracture from a recent fall may report as "disabled"

-This same person may have been fully mobile during the period when screening was offered

- Conversely, a currently mobile person may have been disabled during the screening  period

Recommendation:

Provide clear justification for assuming the relationship between current disability status and past screening behavior

Consider incorporating disability duration or onset timing in the analysis

Address how temporary disabilities might affect the study's conclusions

(This methodological limitation requires convincing clarification to validate the proposed association between disability status and preventive care access.)

Second- The study's assessment period significantly overlaps with COVID-19 restrictions, including the semi-state of emergency that lasted until March 21, 2022. This raises a crucial methodological concern: how can the study accurately measure preventive service utilization when the retrospective period (past 1-2 years) coincides with peak COVID-19 restrictions (2019-2020), during which these services were intentionally limited to prevent virus transmission? This temporal overlap potentially compromises the study's ability to evaluate typical patterns of preventive care access and utilization, challenging the achievement of the study's primary objectives.

Third: for Figure 2: Consider simplifying the functional limitations analysis into two groups:

-Those with one or more functional limitations

-Those without functional limitations

Fourth: For Figure 3: Consider converting the visualization to a detailed table format to better present the multiple subgroups.

C-In the Discussion:

First-Despite pandemic-related restrictions on preventive services, the data reveals an encouraging trend: a decrease in non-participation rates among disabled patients  for general health examinations and cancer screenings compared to 2016 levels. This improvement, achieved even during pandemic constraints, suggests an overall enhancement in Japan's preventive healthcare delivery since 2016. However, to fully contextualize this progress, the study would benefit from incorporating broader health metrics, including quality of life indicators and health outcomes such as life expectancy trends from 2016 onward. These additional indicators would provide a more comprehensive assessment of improvements in preventive care accessibility and delivery within Japan's healthcare system.

Second: Enhance discussion of:

-Pandemic impacts on preventive healthcare access in Japan

-Policy implications based on study findings

-Recommendations for future research examining factors affecting healthcare accessibility among disabled patients

Reviewer 2 Report

Comments and Suggestions for Authors

The purpose of the paper is to identify and describe if and to what extent people with disability in Japan are hindered from the access to preventive health services in comparison with people without disabilities. I find the topic of the paper interesting and important. The study is performed on the basis of secondary data collected in  Comprehensive Survey of Living Conditions in 2022 which seems a valuable source of data as it is a nationwide representative cross-sectional survey. Various preventive health services are considered in the article, particularly general health examinations and cancer screenings (lung, colorectal, gastric, cervical and breast). Functional limitations defined by Washington Group Short Set are considered in some analyses. The authors use descriptive statistics as well as logistic regression approach. The results show that disabilities are factors which should be strongly considered in case of non-participation in preventive health services.

I have particular remarks.

1.The title suggests that the whole paper analyses the relationships between functional limitations and use of General Health Examination and cancer screening. But as I understand, tables 1 and 3 compare the participation of people with disabilities in general with people without disabilities. The tables present category “Disability” and in the Material and Methods section the disability status is defined by the question “Do you need any support or supervision from others because of your disability or declining physical function?” (lines 209-210). If so, tables 1 and 3 treat disability in general not regarding particular functional limitations. Functional limitations are considered in Figure 2 and 3 as well as in supplementary materials – Table S2-1 but in the main text the only information is  “The detail characteristics of participants or nonparticipants in the GHE and the  five cancer screenings are presented in Table S2 in the Supplementary Material.” (lines 283-284). If the title of the paper underlines functional limitations more results concerning this relationships should be described in the text.

  1. In the Abstract in the lines 21-23 “of 23868 study samples (men and women aged 20–74 years) in the CSLC2022, 129–239 people with disabilities did not participate in preventive healthcare services”. It is not clear what you mean by “129-239 people” Why do you provide an interval here?

  1. Lines 134-135 “The statistical data presented in the current study were generated by the authors and thereby differs from publicly available data published by the MHLW.” What do you mean that the statistical data were generated by the authors? How authors can generate statistical data? As you write in line 133 secondary data from CSLC2022 were used for this study. I guess that maybe you selected or transformed the available data from CSLC2022 and if so you should describe that. In statistics data generation is usually understood as a creation of “artificial data” (often following a given distribution) by using computer tools and algorithms, random number generators. Therefore, this fragment of the paper needs to be reworked or you should explain in details what you mean here.

  1. Lines 156-159 “Those who were in special circumstances, such as hospitalization or initialization for more than 3 months during the survey period, or foreigners who were unable to answer the questions written in Japanese were exempted from answering the question.” My concern here is that you use singular form here “from answering the question” – only one questions or all questions from the questionnaire?

  1. Lines 275-277: “The percentage of people with disabilities among the preventive health service participants was 0.80–1.36%. Conversely, the percentage of non-participants was 2.40–3.66, which was approximately three times larger.” In the second part also “among” , i.e. Percentage among, should be used not “percentage of”.

  1. Table one:

„Constant visit to hospitals†”

It seems as at the end there is a kind of asterisk, but not referenced under the table.

But in line 290 there is a statement “No clear definition of the frequency of visits was provided in the questionnaire.” and it is not clear what it refers to, these hospital visits?

  1. Figure 3. In my opinion this figure can mislead the reader if no additional explanations are given. The description in the text referring to the Figure 3 is too laconic to convey the idea of ​​it. As Figure 2 shows, many people have a few functional limitations so accumulating categories which are not mutually exclusive (what I suppose was done in Figure 3) is problematic and needs better explanation. Explain what percentages presents Figure 3 and what information it provides.

  1. Figure 4. In the legend there is “Total popilation” it should be “Total population”

  1. Lines 357-358: “and one-third of the participants had two or more such limitation.” It should be limitations not limitation.

  1. Lines 316-317 “Overall, no obvious differences were observed between CSLC2016 and CSLC2022.” This conclusion is rather drawn from exploratory analysis as I understand. It is probably possible to check it by a statistical test on the base of your data.

Round 2

Reviewer 1 Report

Comments and Suggestions for Authors

Thank you , Well done